# Suitability Study of Using UAVs to Estimate Landfilled Fly Ash Stockpile

**DOI:** 10.3390/s23031242

**Published:** 2023-01-21

**Authors:** Muskan Sharma Kuinkel, Chengyi Zhang, Peng Liu, Sevilay Demirkesen, Khaled Ksaibati

**Affiliations:** 1Department of Civil and Architectural Engineering and Construction Management, University of Wyoming, Laramie, WY 82071, USA; 2Department of Civil Engineering, Gebze Technical University, Cumhuriyet, Gebze 41400, Turkey

**Keywords:** UAVs, stockpile, volumetric analysis

## Abstract

The decrease in fly ash production due to the shift in coal industries toward a green environment has impacted many concrete industries as fly ash is a significant component in cement and concrete. It is critical for concrete industries to identify the availability of fly ash in landfills to meet their demand if the supply decreases. This paper aims to analyze the suitability of UAVs in determining the fly ash stockpile volumes. A laboratory test is performed to validate the proposed UAV method. Then, a real quarry site is selected to demonstrate the suitability in a large scale. The results indicate that the UAVs estimate the most accurate volume of the stockpile when the flight height is about five times the stockpile height. A considerable range of 3.5–5 times the stockpile height is most suitable for quantity takeoff. The findings of this study provide a recommendation for choosing the most appropriate technology for the quantitative estimation of fly ash in existing landfills on a large scale.

## 1. Introduction

The use of fly ash in concrete has been considered an extensive research topic for several years. Fly ashes of moderately low carbon content and moderately high fineness may be replaced for Portland cement in percentages up to 30 in ordinary construction and about 50% for heavy without impairing the qualities of the concrete [1]. With the increased electricity consumption in society, coal consumption has increased significantly, which has increased Coal Fly Ash (CFA) output, a residual material obtained after burning coal. According to a recent survey by the American Coal Ash Association (ACAA), in 2020, roughly 26,512,322 short tons of CFA were produced, with approximately 17,104,493 short tons being utilized [2]. However, due to the increased retirement of coal-fired power plants to comply with environmental emission regulations, fly ash production has been declining yearly. With the increased utilization of the produced coal ash, it is anticipated that there can be shortages in fly ash in the future. Hence, the concrete industry must identify the availability of fly ash disposed of in various storage facilities across the country. Landfilled Fly Ash (LFA) is the unused fly ash dumped in landfills when the CFA supply exceeds demand or the quality does not match the concrete standard. It is critical to identify the quantity of fly ash stored in these landfills to be utilized in the concrete industries to evaluate their suitability to meet the concrete standards. Thus, the first step in this process should be to research the availability of landfilled fly ash in the landfills of coal-fired power stations. An accurate and reliable estimate of the available fly ash stored in the landfills of coal-fired power plants can be a key to survival for concrete industries.

No standard method has been developed to calculate the quantity of fly ash in the existing landfills. Data provided by power plants can help determine the availability. However, it can only be helpful if the power plants have kept a record of past data. In addition, the accuracy of the past data becomes questionable. Hence, it seems practical to thoroughly investigate the number of LFAs by collecting the existing data from the field itself. The conventional methods include surveying by using total stations. The volumetric estimate is based on average-end-area or cross-section methods. However, these methods have imprecision and limitations [3]. These methods are tedious and time-consuming due to the irregularity of shapes and large areas. For example, the average-end-area method generally overestimates the results, and cross-section methods assume that the ground slope is uniform [4].

Due to these shortcomings of conventional methods, developing several innovative engineering technologies and using them in the actual field have become essential. The application of emerging technologies such as 3D scanning, photogrammetry, Building Information Modeling (BIM), and geospatial modeling has been rapidly developing in the field of engineering. For instance, 3D laser scanning can capture an actual image and convert it into a 3D model. This technology is extensively used to estimate the volume of completed works and to determine the site’s progress [5]. BIM can enhance construction safety performance with its collaborative and cooperative environment and information-sharing mechanism [6]. To conduct a volumetric analysis of stockpiles, the three-dimensional (3-D) method is a more prevalent and advanced way [3]. Photogrammetry is a widely used 3-D measurement technique that can be applied to imagery using emerging equipment called Unmanned Aerial Vehicles (UAVs) [7].

A UAV, commonly known as a drone, is an aircraft that is not guided by any human pilot but autonomously by remote control and carries sensors within them. UAVs are widely distributed in surveying [8] construction sites to capture 3D images and determine the progress, estimating the volume of earthwork [3,9,10,11], assessing and inspecting existing structures [4,12] and the environment [4,13]. These UAVs are a type of aerial platform that can acquire digital images and georeferencing. Still, because of their autonomous capabilities and integrated inertial and imaging systems, they are efficient and effective for developing high-resolution digital terrain models (DTMs) and orthoimages. UAV platforms have proven to be an innovative and reliable source of data capture, acquisition, 3-D modeling, image analysis, and assessment [14]. The recent technological advancement has made UAVs capable of estimating the stockpile volume of earthworks on the Earth’s surface.

The UAVs are being utilized in various areas that include but are not limited to architecture, engineering, construction, agriculture and forestry, remote sensing, and 3D mapping and modeling. Abaeaino et al. [15] discovered seven main areas in the Architecture, Engineering, and Construction (AEC) domain where the UAVs are extensively applied: structural and infrastructure inspection, transportation, cultural heritage conservation, city and urban planning, progress monitoring, and post-disaster and construction safety [15]. Based on the scale of the area of interest, several research studies have extended to comparing the suitability of UAVs with other available technologies such as laser scanners, and it was found that UAVs were preferred for larger-scale projects. Rosca et al. [16] presented a comparative study of Terrestrial Laser Scanners (TLSs) and UAVs in the context of feasibility and differences of using both technologies to assess the top of the canopy structure in a tropical forest of Guyana. A statistical analysis of the results showed that UAVs and TLSs are similar, especially in an undisturbed canopy, and different in smoothness and preciseness. UAVs were found to generate smooth data and were less precise than TLSs [6]. Similarly, Cunha et al. [17] investigated the suitability and accuracy of drones and laser scanners in a sizeable outdoor crime scene at a parking lot in Brazil. The study’s findings inclined toward using UAVs rather than Laser Scanners, with the former being more economical and less time-consuming. The existing research on the application of UAVs in civil engineering and construction areas infers that UAVs can be an authentic tool to replace the dependency on satellites and human-crewed vehicles.

Studies have demonstrated a wide range of applications of UAVs in civil engineering and construction. Engineering personnel has been applying UAVs in surveying [9] to determine progress activities [10], estimate earthwork volume and cut/fill quantities [3,11], stockpile volume estimation, and more related topics. However, most of the available research in construction has focused on the application of UAVs in quantity takeoff. Zeng et al. estimated the sediment deposited at the check dams using the UAV photogrammetry techniques in a complex topography [18]. They presented a new method for estimating the sediment silted by check dams with a high-resolution DEM of empty gullies around the check dams captured by UAVs. Various empirical equations with relationships between area and volume were determined in this study. The result showed an error of 12–13% in single check dams and 2–3% in regional check dams. A study by Kim et al. [11] investigated an earthwork volume calculation method for construction sites that monitored earthwork progress at a construction site using UAVs. The study compared a chaining method with a planned plane map based on the average end-area method and Digital Surface Model (DSM) method to calculate the earthwork volume. The DSM method was applied based on the photographs captured by the UAVs. The results showed less error in the calculation performed by DSM compared to the chain-based method, suggesting the suitability of UAVs.

Tamin et al. [19] identified the root-mean-square error and relative error of using a UAVs compared to a TLS in terms of planning, data collection, data processing, and stockpile volume. The RMSE of UAVs was found to be ±0.05 m compared to that of TLSs, which was 0.41 m, and the relative error was 0.002%, with UAVs being more precise than TLSs. Hugenholtz et al. [20] estimated the stockpile volume of gravel in a site in Canada. Two surveys were conducted before and after a part of the stockpile was excavated by two methods, UAVs and Global Positioning System (GPS). The authors compared the volume change based on the haul ticket of the removed gravel from the municipality. There were 2.6% and 3.9% differences in the volume calculated by the two methods before and after excavation, respectively. Further, the results obtained from UAVs DTMs, compared to the haul ticket volume change, were 2.5% higher. All the studies on UAV applications have given positive suggestions on using UAVs in determining the stockpile volume.

Filkin et al. [21] proposed a landfill monitoring approach with the application of UAVs in a landfill made of municipal solid waste (MSW) in Russia. The study compared the suitability of two types of drones, a non-specialized low-cost and a specialized geodetic-class UAV, for quantity takeoff. No significant difference was found between using different UAVs based on errors. However, the landfill Digital Terrain Model (DTM) accuracy was higher in the specialized drone. Only a 1.0% variation was observed in the results from UAVs when compared to the ground survey data. This study has recommended the use of UAVs to monitor an MSW landfill.

Studies have been conducted to analyze the suitability and accuracy of UAVs in estimating the stockpile volume of materials. Existing studies [11,18,19,20] have focused on calculating the stockpile volume of an existing stockpile of some construction project sites or other places, such as a case study of an existing location. There has been a general comparison between different methods such as chaining methods, the average-end area method, UAVs, and laser scanners adopted to calculate the stockpile volume. In addition, these studies have calculated the errors by comparing the results obtained from one method to another [20,22]. This is an indirect measure of the actual volume and might raise uncertainty as there might be various losses or errors in acquiring the data from the respective representative. There have not been many studies conducted where the data are collected from scratch, i.e., laboratory data collection. This study has validated and checked the accuracy of UAVs by collecting the actual data sample by preparing a laboratory site. This research aims to address this gap by calculating and validating the stockpile volume of construction material dumped at a laboratory site. Then, a case study is conducted to determine the quantity at an existing quarry site.

This paper is described in seven different sections. Section 2 briefly describes the flow of tasks conducted in this study. Section 3 gives an insight into the analysis and validation of the laboratory work performed in this research, followed by Section 4, which is a case study analysis demonstrating the findings of the laboratory study into a large-scale quarry site. Finally, Section 5 and Section 6 discuss the significant findings, problems, and challenges incurred in this study and the benefits and impacts of the study.

## 2. Methodology

This section presents the methodology adopted in this study for studying the suitability of UAVs in quantity takeoff. In the laboratory site, a certain known quantity of construction material is dumped as a stockpile, and the photos are acquired with the help of a drone. The image from the sites is processed to generate a 3D point cloud model, and the stockpile volume is calculated. Further, the calculated volume is compared to the actual volume to obtain the errors associated with each case. The error calculated from the laboratory site will imply the suitability of the drone more precisely as the actual volume of the stockpile is known in this case. After validating the laboratory study, a case study is performed at a quarry site based on the validated procedure of the laboratory study to determine the feasibility of a drone to estimate the stockpile volume in a large-scale site. The quarry site already has a stockpile of material whose images are captured with a drone and processed in photogrammetry software to estimate the volume.

### 2.1. Laboratory Analysis

The primary purpose of conducting the laboratory analysis is to verify whether the UAVs accurately estimate the volumes in the actual site. There is no direct way to measure the volume of a stockpile existing at a site. The data used to compare UAV results are generated through indirect means such as a haul ticket [20] or stockpile reports. Hence, a laboratory analysis is required to determine the accuracy of the UAVs. In this study, a laboratory is set-up at a site where a stockpile is created by dumping the materials in the field from scratch.

#### 2.1.1. Capture of Aerial Image

##### Selection of Equipment

A study by Filkin et al. [21] revealed that a low-cost, non-specialized UAV could be feasible in obtaining good results for volume estimation in a relatively small project. This study mainly focused on examining the suitability of UAVs in estimating the stockpile volume. The experimental and case study chosen in this case were smaller projects. Hence, an economical and average UAV was chosen for data capture. DJI Air 2S is a manually operated GPS drone with four antennas, 12 km FHD Transmission, obstacle sensing in four directions, a 1-inch sensor, and 2.4 µm pixels that can capture high-clarity images [23]. The recommended threshold for the flight height is 30 m for safety purposes [24].

##### Site Description

Many factors contribute to the selection of the site location, including accessibility at the site, location, topography, external disturbances, wind, permits, and licenses. The laboratory site selected for the experiment was near Alsop Lake, located in Laramie, Wyoming. The site selection for the experimental study was based on several factors mentioned above. A preliminary site visit was conducted to observe the surroundings, topography, weather, location, and accessibility. The site was a public-access area, free from external disturbances and with no high wind. The test area was a flat surface with no extreme topography. A location image and topography map of the site are shown in Figure 1.

High winds are fatal to the flight of drones [25]. Hence, the weather would be a significant factor that could influence the capture of aerial images in this study. Choosing an appropriate time to collect the data was a critical factor in this project. Thus, being mindful of the weather conditions, the data were collected during the start of fall when there was not much wind and the day would be sunny.

##### Capture Parameters

The drone was flown at various altitudes with various camera angles to capture the best images possible. During the data collection, various factors were considered, including flying altitude, capture angle, and drone position concerning the stockpile. The laboratory site data were collected at three different flight altitudes, 2.5 m, 3 m, and 5 m, above ground level. Table 1 shows the frequency of the images captured for each flight altitude.

##### Length of Capture

DJI Air 2S has a maximum flight time of 30 min only. However, three batteries could capture images for three phases of flight altitude. Hence, the capture length was no more than 30 min for each set of flights, resulting in a total time of capture of 150 min.

#### 2.1.2. Data Analysis and Validation

##### Data Processing

After capturing the aerial image, the next step was to process the image into accurate 3-D Point Cloud data. The captured images were refined based on the quality of the image, height of the flight, and position of the camera and imported into the photogrammetry software to create a 3D Point Cloud Model. After the 3D point cloud was generated, the stockpile volume was calculated to obtain the results.

The image was processed through a professional photogrammetry software, Pix4D Mapper by Pix4D. Currently, a wide variety of photogrammetry software is used to process UAV images. Several research studies [3,26,27,28,29] and industries are extensively using Pix4D mapper to process their data and measure the stockpile volume in their analysis due to its good speed and accuracy [30,31]. This software can transform an aerial image into digital maps and 3D models and accurately measure and inspect mathematical parameters such as areas, distances, and volumes. The software can process an image and deliver them in a variety of outputs models such as Point Clouds, Orthomosaic, Contours, Digital Terrain Models (DTMs), Digital Elevation Models (DEMs), Digital Surface Models (DSMs), Index Maps, Thermal Maps, Reflectance Maps, and a 3D textured mesh. However, at least three images are required to analyze the data in this software.

Point Cloud generation in Pix4D mapper takes place through a machine learning process. The software automatically detects all the common points between the images imported for analysis. The software automatically identifies the coordinate system of the images, with the default being the World Geodetic System (WGS) 84. The initial process involves developing automatic tie points and a 3D model created from each 3D point of the images. After the initial processing, the automatic tie points created will result in a Densified Point Cloud, with the extra tie points created based on the automatic tie points. The final step involves the creation of the desired output model mentioned above. The DSM created will facilitate determining the volume of the image processed. Based on the boundary points of the stockpile given by the user to determine the volume, the software generates a base considering the attitude of each point. Grids are generated based on the Ground Sampling Distance (GSD) spacing, which is the distance between the centers of the pixels or grids on the ground. The software then calculates the volume of each grid, which is given by Equation (1):V = Li ∗ Wi ∗ Hi(1)
where:

Li = Length of each grid;

Wi = Width of each grid;

Hi = Difference between the terrain altitude and base altitude of each grid.

Two different types of volume, the cut volume and the fill volume, are given by the Pix4D mapper. Cut volume is the total volume of the cells when the 3D surface is higher than the base, while fill volume is the total volume of the cells when the 3D surface is lower than the base. The total volume of the surface is the sum of the cut volume and fill volume.
Total Volume = Cut Volume + Fill Volume(2)

This study used only the cut volume as the 3D surface, i.e., the stockpile, was above a flat surface.

#### 2.1.3. Data Validation

The laboratory data were validated with the actual volume to determine the error percentage. The model generated by the software was analyzed from the quality report provided by the software. The stockpile volume was then compared to the actual volume.

### 2.2. Case Study

After the analysis of the experiment conducted at the laboratory site, a standard method of using UAVs to conduct the volumetric analysis was established. Hence, based on the findings of the laboratory analysis, a quarry site was chosen to demonstrate the suitability of UAVs in the actual large-scale field. Following the exact procedure of the laboratory analysis, the aerial images of the stockpile were captured, processed through Pix4D mapper, and the volume estimated.

## 3. Laboratory Analysis

### 3.1. Design of Experiment

For conducting the experiment, the laboratory site was set-up at a public access area at Alsop Lake, Laramie, Wyoming. A less windy and sunny day during the fall was chosen for the experiment. The experiment was set-up during the morning so the wind would not have much effect while the data were collected. Ten bags of paver base sand, each 0.014 m^3^ in volume, were purchased and dumped at the site to create a stockpile of about 1 m in height. The selection of sand as the construction material was based on various reasons. Its maximum dry density ranges from 1000 to 1700 kg/m^3^ and is composed of oxides of silica, alumina, and carbon, which resembles sand [32]. In addition, studies have suggested that fine aggregates such as sand have great potential to replace the use of fly ash in concrete [33]. The sand was placed in a thin plain sheet on the ground to facilitate image processing. The drone was flown at three altitudes over the stockpile to capture the image.

### 3.2. Data Processing

A total of 41 images were captured in total, with 17, 12, and 12 images captured at 2.5 m, 3.5 m, and 5 m altitudes, respectively. The imported images were processed in Pix4D mapper, resulting in a point cloud model representing various forms, as shown in Figure 2a–c.

Boundary points were identified in the point cloud model, and the volume was computed as presented in Figure 3. The base limit was set as triangulated, as this option is recommended for stockpiles whose boundary is visible [34].

The quality of the generated 3D point cloud model primarily depends on the captured images. The level of coverage of the captured images while processing the point cloud model is one of the factors responsible for a good-quality 3D point cloud model. The quality of the generated 3D point cloud model primarily depends on the captured images. The level of coverage of the captured images while processing the point cloud model is one of the factors responsible for a good-quality 3D point cloud model. More images give higher coverage, and higher coverage gives higher accuracy. Hence, the image processing was conducted in two phases to check the quality of the results. First, all the images captured at each altitude were imported and processed. The volume was calculated for each set of flight altitudes. Then, some images were deleted, and the resulting number of images was processed to generate the point cloud model and estimate the volume. The average ground resolution of the pixels of images was 0.22 cm/0.09 inch. The photogrammetric triangulation RMS error was 0.27%, 0.21%, and 0.68% in X, Y, and Z directions, respectively. A detailed representation of the number of images processed at each phase of analysis is presented in Table 2.

### 3.3. Data Validation

The results obtained from the volumetric analysis conducted at each phase were validated with the actual volume of the stockpile, which was 0.14 m^3^ for this case, as 10 bags of sand, with each bag being 0.014 m^3^ in volume, were used. The errors were calculated for each data analysis phase: (1) when all the captured images are processed and (2) when some images are deleted and then processed. An average of the errors determined in the two phases was estimated. The calculations of errors are presented in Table 3.

The results show that the data captured from the highest altitude gave the lowest error. In addition, the percentage of error decreased as the flight altitude increased. This notable finding directs to the conclusion that flight altitude is a significant factor influencing the volume calculation’s accuracy. A higher difference between the top elevation of the stockpile and the drone elevation gives more accurate results. The estimated volume calculated for all the images imported during phase I of the data analysis was the same when the flight altitude was 3.5 m and 5 m. As the height of the stockpile was 1 m, it can be inferred that the height of the flight can be in the range of 3.5 to 5 times the height of the stockpile to improve the accuracy of volume estimation.

It was also observed that the error percentage was less when all the captured images were processed at the height 3.5 m and 5 m. Hence, it can be concluded that the volume estimation is the most accurate when all the captured images are processed to generate the 3D point cloud model, and when the altitude is 5 times the height of the stockpile.

## 4. Case Study

Based on the lab findings, a case study was conducted to assess the suitability of UAVs in an actual large-scale site. The quantity takeoff at a fly ash site could not be performed, due to several reasons, including restricted accessibility permits, prohibition of drones in the landfill sites, and safety concerns because most of the sites had hauling operations in progress. A fly ash site is generally an industrial site and is large in scale. Hence, to assess the suitability of UAVs in determining the stockpile volume of fly ash, a location that could be similar to the fly ash site was selected, such as a quarry site. The quarry site selection was based on similar factors as in the case of the laboratory site. Of all the possible locations, Harriman Quarry Site, located about 30 miles west of Laramie and operated by SIMON Contractors, was chosen for the case study experiment. The site location is shown in Figure 4. An accessibility permit was acquired from the manager of the quarry site. The site had numerous stockpiles of sand, aggregate, granites, and other minerals dumped within the area. For this study, the sand stockpile was selected for safety reasons and ease of access. The quarry site had a flat topography, where leveling operations were performed before the dumping of the stockpile.

### 4.1. Capture of Aerial Image

About 65 pictures were collected as the stockpile was massive, about 7 m in height. Based on the findings from the laboratory analysis, the drone’s altitude was maintained at 25 m from the ground level. A higher flight height could not be achieved, as (1) the height limitation, as per the drone specification, was 30 m from the ground level for safety purposes; (2) there was a very high wind that could have caused distortions. However, the flight height was 3.6 times the height of the stockpile, which is also acceptable as per the laboratory results. Around 65 images were captured from different angles and positions to obtain the best results. Figure 5 represents the primary position and path of the drone during the capture process.

The frequency of images captured at each of these positions is presented in Table 4.

### 4.2. Image Processing

As concluded from the laboratory analysis, all 65 captured images were imported to create the point cloud model. The average ground resolution of the pixels of images was 0.9 cm/0.36 inch. The photogrammetric triangulation RMS error was 0.31%, 0.37%, and 3.1% in X, Y, and Z directions, respectively. The imported images were processed in Pix4D mapper, resulting in a point cloud model representing various visual forms, as shown in Figure 6a–d.

### 4.3. Volumetric Analysis

Figure 7 shows a representation of volumetric analysis, and Table 5 provides the results obtained from the same.

The actual volume used for comparing the calculated results was taken a couple of days earlier (10 October 2022) than when the case study was performed (14 October 2022). The data were acquired from the stockpile report provided by the site manager for the month the case study was performed. The quarry site generates the stockpile report every time a haul operation is performed at the stockpile. According to the stockpile reports, no haul operations were performed from 10 October 2022 until 31 October 2022. Hence, the actual volume from the stockpile report generated on 10 October 2022 was chosen in the case study.

## 5. Discussion

Alleviating fly ash production from coal power plants has become a severe issue for the concrete industries. Professionals are looking for various ways and alternatives to tackle the scarcity of fly ash in concrete if any arise. They need to estimate the quantity of fly ash stored in landfills. It was crucial to check if UAVs could give an accurate result if a proper procedure is adopted. Hence, the suitability analysis was conducted in a real laboratory site with the stockpile created from scratch. The results showed that a UAV could best estimate the volume with the highest accuracy when flown at a higher altitude with respect to the height of the stockpile. This might be due to the direct linear relationship between the GSD and flight altitude. A UAV flown at a higher altitude covers more area of the image and the GSD becomes higher, which makes the image less detailed, which might be a reason for the higher accuracy. The results depict that the flight range can be 3.5 to 5 times the stockpile height to obtain an accurate result. Although the results were validated from the actual data, it is also essential to know if the point cloud model generated met the quality standards because the volume estimation entirely depends on the point cloud model generated by the software. For instance, there is a difference between the point cloud model generated using all the captured images and only one image per capture position. Using all the images gives a better point cloud model as more tie points to connect the images are created when there are more images. However, when there are insufficient images, enough tie points to connect the images cannot be created, resulting in some images being uncalibrated, resulting in holes and distortions in the point cloud model. Hence, the image should be captured from all directions covering the entire site to avoid the images being uncalibrated. The quality of the point cloud model generated in the Pix4D mapper can be checked through a quality report. The quality report can check a series of parameters in the point cloud model [35]. The following conditions were verified to check whether the generated model has good quality: (1) The percentage of images calibrated is near or equal to 100%. (2) The relative difference between initial and optimized internal camera parameters is below 5%. (3) The orthomosaic does not contain holes and distortions. (4) Overlap: A green color indicates good quality. (4) The image coordinate system is correct. (5) For the absolute camera position uncertainty, the sigma is smaller than the mean. The interpretation of the quality reports for the point cloud model generated at the laboratory and case study site is presented in Table 6.

### 5.1. Challenges and Limitations

This research has provided a guideline for researchers and industry professionals in stockpile volume estimation. The largest challenge during this research was to fly the drone at a high altitude in the wind, which caused distortions. In addition, maintaining the direction of the camera and the distance between the position of each capture was challenging because the improper orientation of the flight positions and insufficient coverage of the site while capturing could result in uncalibrated images.

One limitation of this study was that only one drone was used to conduct the research. The drone used to capture the aerial image had a maximum flight height limitation of 30 m for safety purposes. Hence, the drone could not be flown at higher levels to test if the results were more accurate at higher heights. The errors were calculated in this study based on the results obtained from the software compared with the actual data. The generated errors can be due to triangulation error, camera distortions, Ground Sampling Distance (GSD), and others. The error source and category can be investigated further and considered for future studies. In addition, there is the fact that the test material used at the laboratory site, i.e., the sand, is different from the target material, i.e., the fly ash. The density of these materials is slightly different, resulting in different stockpile shapes. Hence, the calculation of volume estimation might differ for different materials and shapes of the stockpile. Further, this study only considered flight height as a parameter to govern the capture process. Other factors such as the capture angle and different number of processed images can be considered to investigate the quantity takeoff. This gives further research directions for the studies conducted in stockpile volume estimation.

### 5.2. Benefits and Impact

The findings of this study will benefit the concrete industry professionals and researchers: (1) to estimate the available fly ash in the landfill by adopting a standard procedure of volume estimation, (2) to further investigate other similar technologies that can estimate the stockpile volume, and (3) to look on the benefits of the UAVs in comparison to the traditional methods of volume estimation.

Concrete industry professionals will perceive a method for estimating the quantity of fly ash in the existing landfills of coal-fired power plants. This will help them investigate the quality of the landfilled fly ash and check whether the LFAs meet the standard concrete regulations. In addition, they can decide if the fly ash needs any beneficiation. A proper estimation of the availability of LFAs will also help them to prepare for future shortages. Alternatives to fly ash suitable in concrete can be investigated in the future.

Researchers can investigate the suitability of similar technologies that facilitate quantity takeoff. A feasibility analysis can be conducted in a detailed manner and the cost incurred from each method can be analyzed to determine the most suitable technology.

UAVs are one of the most burgeoning emerging technologies in construction nowadays. In contrast to traditional surveying and volume estimation methods, UAVs perform better, with ease, less time, and less money. The traditional measurement methods carried out by total stations and levels are costly and time-consuming. These methods incur high equipment costs, labor costs, maintenance costs, more human resources, and less safety. They cannot estimate the volume of irregular shapes accurately, as they adopt manual methods of estimating the volume such as the trapezoidal, planned plane, and average end-area [11,36]. On the other hand, technological advancement has made UAVs an easy-to-handle tool that requires fewer human resources, cheaper costs, less time, and is safer. Hence, this research suggests that UAVs are an efficient technology for effectively estimating the quantity of a stockpile on a large scale.

## 6. Conclusions

This study has demonstrated the suitability of UAVs to carry out quantity takeoff and determine the stockpile volume at laboratory and case study sites. This paper aimed to study whether UAVs can be beneficial in estimating the quantity of landfilled fly ash in coal power plants or any other industrial sites. The analysis was conducted at a laboratory site, and the results were validated. The findings of the laboratory analysis suggested that UAVs can estimate the quantity with reasonable accuracy. It was observed that the accuracy of the drone in calculating the volume of the stockpile at the laboratory site decreases as the flight elevation of the drone increases. This finding has led to the conclusion that volume estimation can be achieved better when the drone is flown at a higher altitude with respect to the height of the stockpile. It can also be inferred that a better-quality point cloud model can be generated that improves accuracy in volume estimation when the flight height is 3.5–5 times the stockpile height. With this validation, this research moved toward a case study site more significant in scale than the laboratory site. To follow the safety recommendations on the drone’s maximum height and avoid wind distortions, the drone’s altitude was fixed at 25 m from the ground level, about 3.6 times the stockpile height, and the volume was estimated successfully. This study also explored an existing feature in the software: the quality report. The quality report provided validation and quality check of the point cloud model generated during the image processing. All the criteria that were checked in the quality report met the standard requirements for a good-quality dataset by the software for both cases. This infers that the point cloud model generated is accurate and has good-quality data, giving an accurate volume estimation. Hence, it can be concluded that UAVs are suitable for estimating the volumes of fly ash in a landfill.

## Figures and Tables

**Figure 1 sensors-23-01242-f001:**
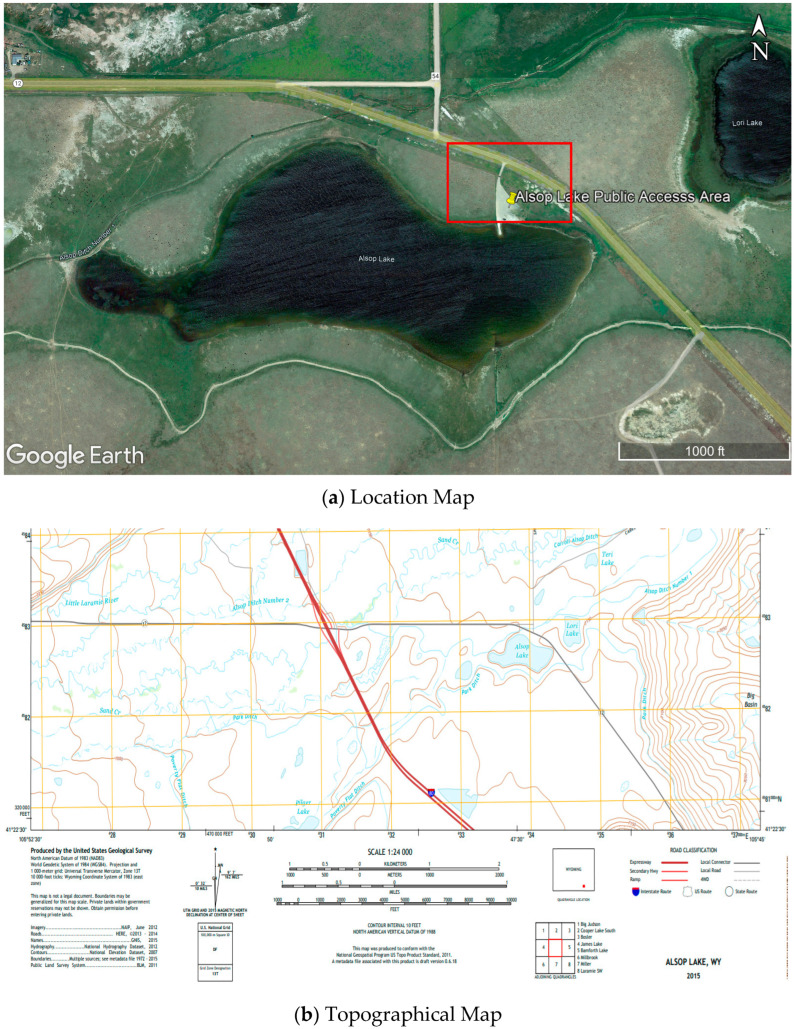
Location and Topographical representation of the site: (**a**) Location Map (Imagery ©2023 CNES/Airbus, Maxar Technologies, USDA/FPAC/GEO, Map data ©2023, reproduced with permission from Google, 2023); (**b**) Topographical Map (reproduced with permission from United States Geological Survey (USGS), 2015) Weather Conditions.

**Figure 2 sensors-23-01242-f002:**
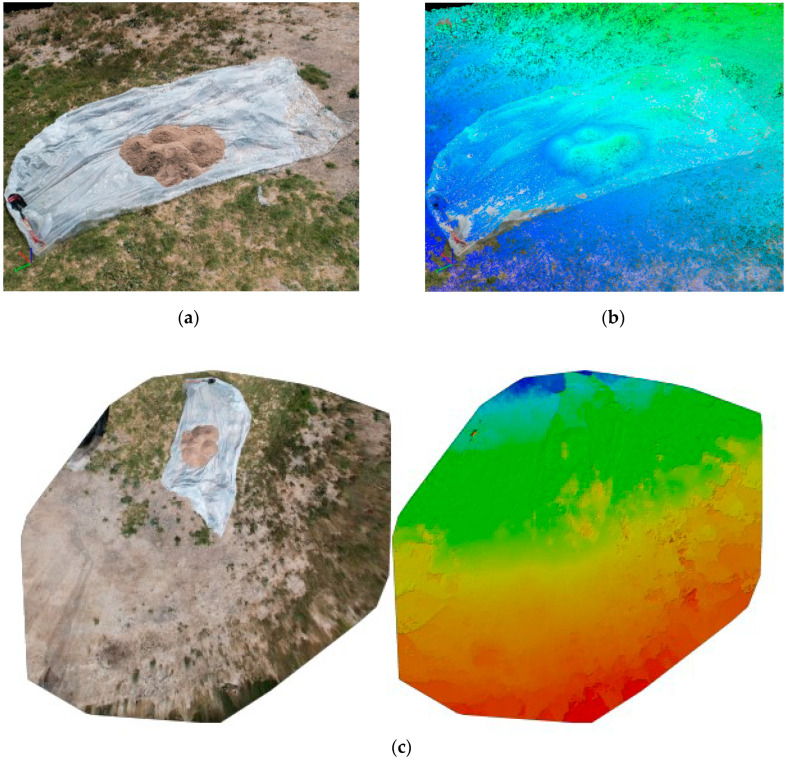
Representation of the generated point cloud model of the laboratory site: (**a**) Point Cloud Model; (**b**) Altitude Model; (**c**) Orthomosaic Model with its corresponding Digital Surface Model (DSM).

**Figure 3 sensors-23-01242-f003:**
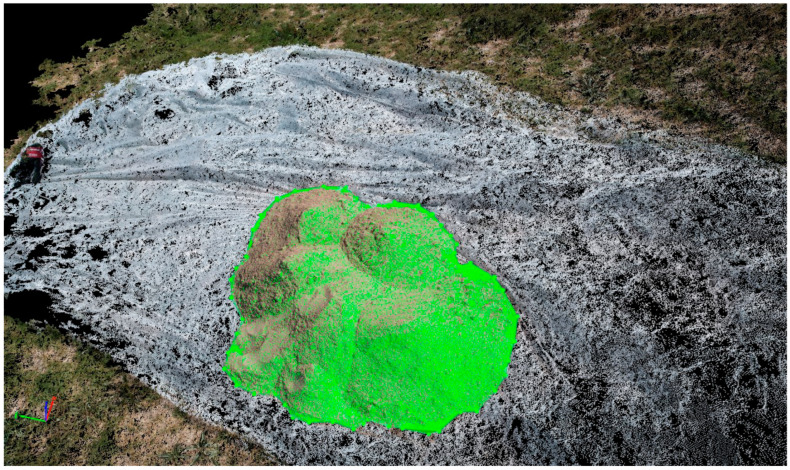
Stockpile Volume Calculation in point cloud model at Laboratory Site.

**Figure 4 sensors-23-01242-f004:**
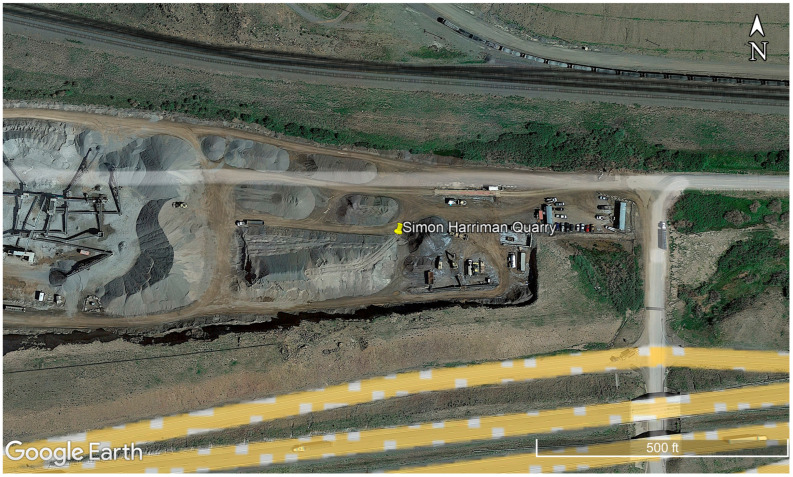
Location of the quarry site (Imagery ©2023 CNES/Airbus, Maxar Technologies, Map data ©2023 Google, reproduced with permission from Google, 2023).

**Figure 5 sensors-23-01242-f005:**
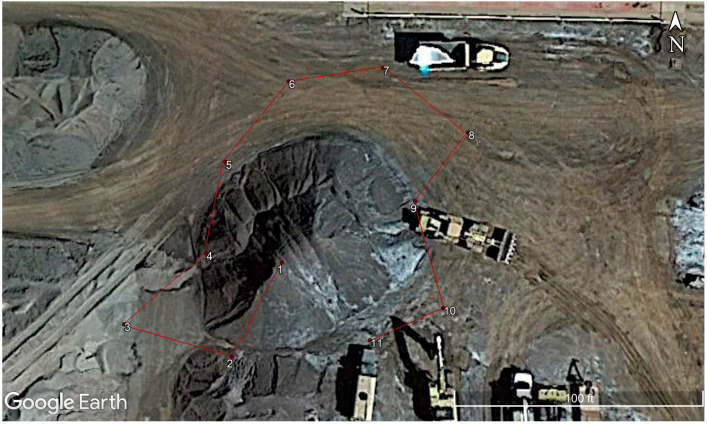
Positions of the drone during capture starting from point 1 till point 11 (Map data ©2023, reproduced with permission from Google, 2023).

**Figure 6 sensors-23-01242-f006:**
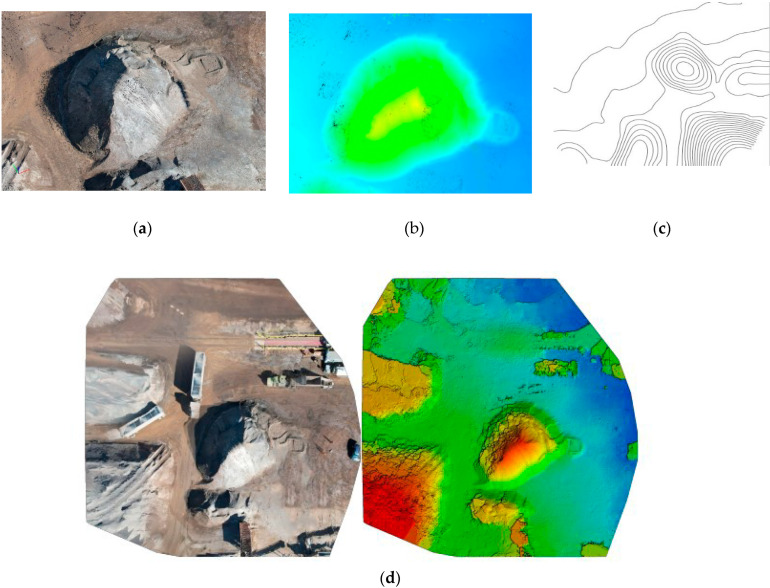
Representation of the generated point cloud model of the quarry site (**a**) Point Cloud Model; (**b**) Altitude Model; (**c**) Contour; (**d**) Orthomosaic Model with its corresponding Digital Surface Model (DSM).

**Figure 7 sensors-23-01242-f007:**
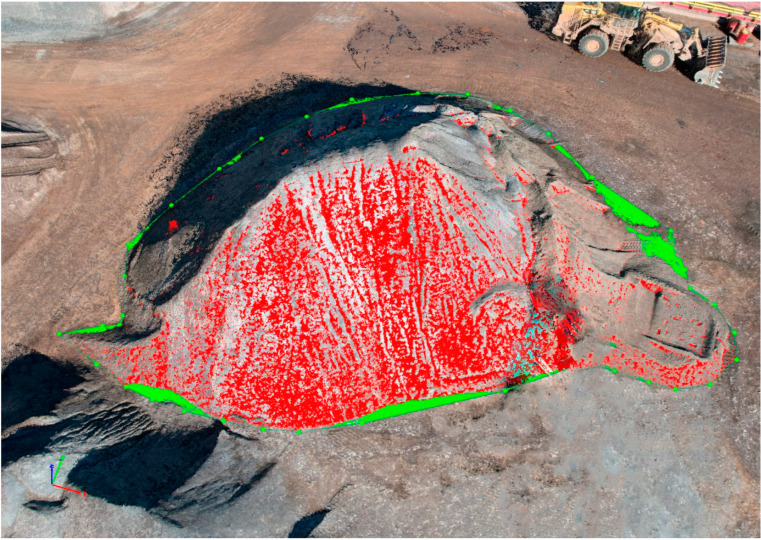
Stockpile Volume Calculation in point cloud model at Quarry Site.

**Table 1 sensors-23-01242-t001:** Number of images captured for each flight height.

Altitude (Meters)	Frequency
2.5	17
3	12
5	12
**Total**	**41**

**Table 2 sensors-23-01242-t002:** Frequency of images captured at each phase.

Flight Altitude (m)	Total Number of Images	Number of Images Processed in Phase I	Number of Images Processed in Phase II
2.5	17	17	6
3.5	12	12	6
5	12	12	7
Total	41	41	19

**Table 3 sensors-23-01242-t003:** Validation of Volume Calculations from Laboratory Analysis.

Phase	Flight Altitude (m)	Number of Images	Calculated Volume (m^3^)	Actual Volume (m^3^)	Error	Average Error
I	2.5	17	0.33	0.14	135%	89%
II		6	0.2		43%	
I	3.5	12	0.12	0.14	14%	21.5%
II		6	0.10		29%	
I	5	12	0.12	0.14	14%	17.5%
II		7	0.11		21%	

**Table 4 sensors-23-01242-t004:** Frequency of images captured at each location group.

Location	Frequency
L1	6
L2	4
L3	2
L4	2
L5	7
L6	2
L7	3
L8	15
L9	5
L10	9
L11	15
Total	65

**Table 5 sensors-23-01242-t005:** Results of volume calculation from the case study.

Calculated Volume (m^3^)	Actual Volume (m^3^)
1453.22	1649.14

**Table 6 sensors-23-01242-t006:** Results from the Quality Report of Laboratory and Quarry Site.

Quality Report Requirement	Laboratory Study	Case Study
Image Calibration = 100%	100%	100%
The relative difference between initial and optimized internal camera parameters < 5%	0.81%	0.63%
Quality of Orthomosaic	No holes and distortions	No holes and distortions
Overlap quality	Good 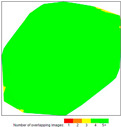	Good 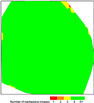
Image Coordinate System	Correct	Correct
Absolute camera position uncertainty (Sigma < Mean)	Sigma < Mean	Sigma < Mean

## Data Availability

The data presented in this study are available on request from the corresponding author.

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
