# Peer review of "Suitability Study of Using UAVs to Estimate Landfilled Fly Ash Stockpile"

_sensors, 2023, doi:10.3390/s23031242_

Round 1

Reviewer 1 Report

This is an useful experiment for estimating volumes of stockpiles using UAVs. The manuscript can be improved in various aspects. 

Line 35: CFA should be defined the first time it is mentioned.

Lines 86-90: What are the methods? What are the comparision results? 

Literature review of the UAV-based methods should be focused on their applications relevant to this study. Section 2.1 is irrelevant. 

Related to the note above, a condensed version of literature review of the existing UAV-based methods should be integrated into the introduction section. A saparete literature review section is not necessay. 

Figure 1 is not necessary

Lines 200-202 can be removed.

Lines 307-311 can be removed.

Section 4 contains much redundancy with section 3. It should be integrated into section 3. 

A quarry site instead of a fly ash site was used for case study. How does the result transfer to a fly ash site?

Line 429: "lowest accuracy" should be "highest accuracy"?

Besides flight height, are there recommendations on other specifications? For example, how many images are needed for different sizes of stockpiles?

Author Response

The authors thank the reviewers for their constructive comments and suggestions. We carefully considered the comments and made the necessary modifications and improvements in the revised version of our manuscript. We think the revised manuscript reads much better than the original version. We hope that the revised paper meets the concerns raised by the reviewers.

Reviewer 2 Report

The methods implemented happen enough effective and operative to be advantageously used in the sectors of concrete production, but in my opinion, some aspects about the contribution of the trend of the real surface beneath the piles on the final volume estimates needs to be better evaluated and accuracy assessment should be better supported, explained and discussed. Your final consideration about the best results obtained using the less detailed images should be better supported and justified taking into account also errors increase linked to GSD, with a brief discussion of this specific aspect.

Globally the paper requires clearer explanation and discussion of the following aspects: 

·         the evaluation of the different geometries (not always flat) of the real surfaces beneath the flying ash stockpiles should be better accounted taking into account of its variability and its impact on the obtained results;

·         a better validation should be provided by considering also the errors arising from the estimate of stock pile base contour and the mean GSD of mages taken in the flights at different high.

Author Response

The authors thank the reviewers for their constructive comments and suggestions. We carefully considered the comments and made the necessary modifications and improvements in the revised version of our manuscript. We think the revised manuscript reads much better than the original version. We hope that the revised paper meets the concerns raised by the reviewers. Please see the attachment.

Round 2

Reviewer 1 Report

The revision addressed some of my comments and suggestions. But I keep my opinion on the following two points:

1. There does not need to be a separate Literature Review section. Literature review is an essential task that any researcher should conduct during a project. But in writing, it does not need to be a separate section. In a paper, the purpose of literature review is to give readers the necessary context and background. So it should be incorporated into the introduction section. 

2. Figure 1 is not necessary. Again, when conducting a research project, it is necessary to go through these steps. But a paper is different. The readers do not need to know you did literature review as the first step (that was the default for any research project). A flow chart should be used to outline your methodology of fly-ash pile volume estimation, but not how you conducted the project. So if you insist on having a flow chart, it should start from image collection, and then go through the algorithms for image processing, and end with validation. The "Case Study" and "Conclusion and discussion" parts of the flow chart are not necessary, either. These are sections of a paper presentation and not parts of a methodology flow chart. 
